# N-Palmitoyl Serinol Stimulates Ceramide Production through a CB1-Dependent Mechanism in In Vitro Model of Skin Inflammation

**DOI:** 10.3390/ijms22158302

**Published:** 2021-08-02

**Authors:** Kyong-Oh Shin, Sungeun Kim, Byeong Deog Park, Yoshikazu Uchida, Kyungho Park

**Affiliations:** 1Convergence Program of Material Science for Medicine and Pharmaceutics, Department of Food Science and Nutrition, Hallym University, Chuncheon 24252, Korea; 0194768809@hanmail.net (K.-O.S.); sungeuna27@naver.com (S.K.); 2The Korean Institute of Nutrition, Hallym University, Chuncheon 24252, Korea; 3LaSS Lipid Institute (LLI), LaSS Inc., Chuncheon 24252, Korea; 4Dr. Raymond Laboratories Inc., Englewood Cliffs, NJ 07632, USA; bdpark03@gmail.com; 5Veterans Affairs Medical Center, Department of Dermatology, School of Medicine, Northern California Institute for Research and Education, University of California, San Francisco, CA 94143, USA; Yoshikazu.Uchida@ucsf.edu

**Keywords:** ceramide, epidermal barrier function, endocannabinoid system, endcannabinoid receptor, skin inflammation

## Abstract

Ceramides, a class of sphingolipids containing a backbone of sphingoid base, are the most important and effective structural component for the formation of the epidermal permeability barrier. While ceramides comprise approximately 50% of the epidermal lipid content by mass, the content is substantially decreased in certain inflammatory skin diseases, such as atopic dermatitis (AD), causing improper barrier function. It is widely accepted that the endocannabinoid system (ECS) can modulate a number of biological responses in the central nerve system, prior studies revealed that activation of endocannabinoid receptor CB1, a key component of ECS, triggers the generation of ceramides that mediate neuronal cell fate. However, as the impact of ECS on the production of epidermal ceramide has not been studied, we here investigated whether the ECS stimulates the generation of epidermal ceramides in an IL-4-treated in vitro model of skin inflammation using N-palmitoyl serinol (PS), an analog of the endocannabinoid N-palmitoyl ethanolamine. Accordingly, an IL-4-mediated decrease in cellular ceramide levels was significantly stimulated in human epidermal keratinocytes (KC) following PS treatment through both de novo ceramide synthesis- and sphingomyelin hydrolysis-pathways. Importantly, PS selectively increases ceramides with long-chain fatty acids (FAs) (C22–C24), which mainly account for the formation of the epidermal barrier, through activation of ceramide synthase (CerS) 2 and Cer3 in IL-4-mediated inflamed KC. Furthermore, blockade of cannabinoid receptor CB1 activation by AM-251 failed to stimulate the production of total ceramide as well as long-chain ceramides in response to PS. These studies demonstrate that an analog of endocannabinoid, PS, stimulates the generation of specific ceramide species as well as the total amount of ceramides via the endocannabinoid receptor CB1-dependent mechanism, thereby resulting in the enhancement of epidermal permeability barrier function.

## 1. Introduction

Human skin is composed of two major layers, the epidermis and, dermis, and each layer exhibits unique structural and physiological properties [1,2]. Because the outer layer epidermis is positioned at the interface with the environment, it mainly functions as a barrier to protect the body from harmful substances, such as UV irradiation, mechanical damage, and microorganisms [1,2]. A number of factors, e.g., structural and junctional proteins, lipids, and transcription factors, participate in the formation of the epidermal barrier [3]. While, in particular, cutaneous lipids serve as a key constituent to forming an optimal epidermal barrier, ceramides comprise approximately 50% of the intercellular lipid content by mass and signal to modulate multiple cellular functions, e.g., cellular proliferation, differentiation, and apoptosis. Alterations in the epidermal ceramide content have been characterized in certain inflammatory skin diseases, including atopic dermatitis (AD) [4]. In AD skin lesions, the content of ceramides containing long-chain fatty acids (FAs) (22–26 carbons in length) is significantly reduced, whereas ceramides with short-chain FAs (<20 carbons in length) are conversely elevated, leading to compromised barrier integrity [4].

It is widely accepted that the endocannabinoid system (ECS) is involved in multiple biological responses via the Gi/o protein-coupled receptor, known as endocannabinoid receptor CB1- and/or CB2-dependent mechanisms in different tissues, including skin, as follows [5]: (1) the activation of CB1 or CB2 increases endocannabinoid levels by inhibiting fatty acid amide hydrolase or adenylyl cyclase, which in turn activates intracellular kinases such as extracellular-signal-regulated kinase (ERK); (2) CB1/2 modulates certain ion channels, i.e., inhibition of N- and P/Q-type voltage-sensitive calcium channels, and activation of potassium channels [6]. In addition to these established G-protein-coupled events, recent studies revealed that activation of CB1 is tightly associated with the generation of cellular ceramides [7,8]. In the skin, the ECS is profoundly involved in the regulation of epidermal homeostasis, e.g., keratinocyte (KC) proliferation and differentiation. Activation of CB1 stimulates DNA methylation through mitogen-activated protein kinase-dependent pathways, resulting in the suppression of keratinocyte differentiation [9,10]. In addition, CB1/2-coupled signaling mechanisms in the modulation of epidermal KC proliferation have been reported [9].

However, even though ceramides are a key component in the formation of the epidermal barrier and serve as signaling modulators to regulate cellular functions, the impact of CB1 on the production of epidermal ceramides has not been studied. Thus, we assessed whether CB1 activation could stimulate ceramide production in inflamed cellular conditions, in which ceramides are significantly diminished. Here, we show that an IL-4-mediated decrease in overall ceramide levels was significantly stimulated in human KC following N-palmitoyl serinol (PS), an analog of the endocannabinoid N-palmitoyl ethanolamine, through increased activities of ceramide synthetic enzymes. Furthermore, PS selectively increases ceramide levels with long-chain FAs (22–26 carbons in length). Finally, we demonstrate that the activation of CB1 accounts for the PS-mediated increase in the content of total ceramide, as well as ceramides containing long-chain FAs. These studies illuminate that endocannabinoid-mediated activation of CB1 modulates the epidermal ceramide profile, contributing to the improvement of the epidermal barrier function of inflammatory skin diseases in which ceramides are diminished, such as AD.

## 2. Results

### 2.1. Change in Overall Ceramide Production in Human Keratinocytes (KC) in Response to N-Palmitoyl Serinol (PS)

Prior studies demonstrated that cannabinoids, e.g., endocannabinoids, phyto-cannabinoids, or synthetic cannabinoids, could modulate cellular ceramide production through activation of a specific cannabinoid receptor, CB1, in different cell types or tissues [8]. However, the impact of cannabinoids on the generation of cutaneous ceramide has yet been determined. In order to investigate whether cannabinoids regulate ceramide generation in the skin, we utilized N-palmitoyl serinol (PS), an analog of the endocannabinoid N-palmitoyl ethanolamine (PEA) (Figure 1A). Human epidermal keratinocytes (KC) were treated with PS at various concentrations (10–200 μM) for different periods—0.5, 1, 2, 3, 4, or 24 h. As shown in Figure 1B, cell survival was suppressed in a dose-dependent manner. Because a significant decrease in cell viability by PS treatment was found at a concentration of ≥50 μM, we employed PS concentrations of ≤25 μM for the studies described below.

We next determined whether exogenous PS stimulates ceramide production in human KC. LC-MS/MS analysis revealed a significant increase in total ceramide content in cells after 4 h of exposure to PS (Figure 1C). Whereas ceramide levels gradually returned to baseline in cells in response to PS treatment for ≥4 h (Figure 1C). These results indicate that PS differentially modulates the production of epidermal ceramide upon incubation period.

### 2.2. PS Have an Abrogating Effect on the Ceramide Deficiency Caused by IL-4 Treatment in Cultured Human KC

It is well known that type 2 T helper (Th) cytokines and interleukin (IL)-4, significantly attenuate ceramide levels in the skin [11]. We next investigated whether PS could rebound ceramide production in cells in which ceramide levels are significantly diminished by IL-4 treatment. As expected, our lipid analysis revealed that treatment with IL-4 led to a significant decrease in total ceramide contents (≈30% of the untreated control) at a concentration of 50 ng/mL after 24 h without inducing cell death (Figure 2). Whereas an IL-4-mediated decrease in ceramide was significantly stimulated in cells following PS treatment (Figure 2). These results indicate that PS stimulates epidermal ceramide production in an in vitro model of Th2 type cytokine (IL-4)-mediated skin inflammation, where total ceramide levels are decreased.

### 2.3. Both Do Novo- and Sphingomyelin Hydrolysis-Pathways Occurs in PS-Mediated Ceramide Synthesis in IL-4 Pretreated KC

Epidermal ceramides can be produced by the action of several distinct enzymes, e.g., serine palmitoyltransferase (SPT), ceramide synthases (CerSs), or sphingomyelinases (SMases), which are involved in either ceramide de novo synthesis or sphingomyelin (SM) hydrolysis pathways, respectively [12]. To determine which enzymes were responsible for a PS-mediated increase in ceramide, we measured the activity of those enzymes associated with ceramide production. In IL-4 untreated cells, PS increased the activity of SPT, but the activity of CerSs and SMases were not altered (Figure 3A–E). Whereas an IL-4-mediated decrease in the activities of all enzymes (SPT, CerS2/3, and SMase) measured were significantly elevated in cells following PS treatment (Figure 3A–E). These results indicated that PS differently modulates ceramide production in normal (unstressed)- or stressed-cellular conditions.

### 2.4. PS Particularly Increases Ceramides Containing Long-Chain Fatty Acids in IL-4-Mediated Inflamed KC

Because prior studies demonstrated that ceramides containing long-chain fatty acids (FAs) with 22–26 carbons in length are generated by CerS2 and CerS3 during de novo synthesis [13], we further determined the carbon chain lengths of FA of ceramides which were enhanced by PS. Consistent with prior findings [14], we showed that treatment with IL-4 significantly reduces the contents of ceramides with 22–26 carbon-containing FAs in cells (Figure 4A–E). Whereas PS particularly stimulates ceramides with long-chain FAs, such as C22, C24, C24:1. C26, C26:1, which are individual species tightly regulated by CerS2/3, in cells pretreated with or without IL-4 (Figure 4A–E). On the other hand, overall levels in ceramides containing short-chain FAs (≤20) were moderately altered or even not changed under these conditions in contrast with long-chain ceramides (Appendix A). These results indicate that PS stimulates long-chain FAs-containing ceramides by upregulating CerS2/3 activity in IL-4-mediated inflamed KC.

### 2.5. PS Stimulates Ceramide Production through a CB1-Dependent Mechanism in IL-4-Mediated Inflamed KC

To investigate the downstream mechanism responsible for PS-mediated ceramide synthesis, we next assessed the potential involvement of a CB1 cannabinoid receptor-dependent pathway. Again, an IL-4-mediated decrease in overall ceramide production was stimulated in cells following PS treatment (Figure 5). However, the blockade of CB1 activation by a specific pharmacological inhibitor of CB1, AM-251, significantly reduced PS-mediated stimulation of total ceramide (Figure 5), as well as specific ceramide species containing long-chain FAs (C22–C26) in IL-4, pretreated KC (Figure 6A–E), while the inhibitor alone did not alter ceramide production (Figure 6). These results revealed that PS-mediated CB1 activation accounts for the PS-induced stimulation of overall ceramides in IL-4-mediated inflamed KC.

## 3. Discussion

The outermost layer of the epidermis, stratum corneum (SC), is composed of corneocytes, which are embedded in extracellular lipids, including ceramides. SC functions as an epidermal permeability barrier to defend our body against external threats, such as UV irradiation and pathogens [12]. Ceramides, a class of sphingolipids containing a backbone of a sphingoid base (or referred to as sphingosine) that is linked to a fatty acid (FA) via an amide bond, are the predominant lipid comprising approximately 50% of the SC lipid content by mass, and the most effective structural component for the formation of the epidermal permeability barrier [12]. Ceramides are produced in the skin through two major pathways [13]: (i) the de novo synthesis pathway that is initiated by serine palmitoyltransferase (SPT) and catalyzes the condensation of serine and palmitoyl-CoA to produce 3-ketosphiganine, which is further metabolized to sphinganine. Sphinganine is then amide-linked (N-acylated) by ceramide synthases (CerSs) to generate ceramides with different acyl chain lengths; and (ii) the sphingomyelinase (SMase)-dependent hydrolysis of sphingomyelin, a key component of the plasma membrane. In the present study, we show that N-palmitoyl serinol (PS) differently regulates ceramide generation in different cellular conditions, i.e., stressed/diseased condition vs. un-stressed (normal) condition. Because an IL-4-mediated decrease in ceramides could cause abnormal barrier functions—induction of excessive transepidermal water loss, and an increased risk of pathogenic infection—PS likely accelerates the enhancement of the skin barrier function by a prompt increase in ceramide content due to the SM hydrolysis pathway, through activation of SMases in the plasma membrane as well as the SPT/CerS-mediated de novo synthesis pathway under stressed conditions, compared with un-stressed conditions, in which the SPT-mediated de novo synthesis pathway is only operated.

Prior studies have demonstrated that epidermal ceramide levels are altered (increased or decreased) in certain skin inflammatory environments, such as atopic dermatitis (AD) [15]. Ceramide levels are significantly reduced in the lesional skin of patients with AD, a chronic inflammatory skin disease in which the Th2 cytokines, e.g., IL-4 and IL-13, are key drivers involved with the underlying inflammatory process [16]. In particular, ceramides containing long-chain FAs (22–26 carbons in length) are significantly decreased, while ceramides with short-chain FAs (<20 carbons in length) are, conversely, increased in the lesional skin of patients with AD [17,18,19]. Of six CerS isoforms, both CerS2 and CerS3 are expressed at high levels in the skin and have been shown to synthesize the longer ceramide species; whereas other isoforms of CerS are associated with the production of ceramide species containing the shorter FAs [17,18]. Consistent with these findings, PS selectively activates CerS2 and CerS3 to elevate ceramides with 22–26 carbon-containing FAs, which are particularly responsible for the formation of the epidermal permeability barrier in both un-stressed or stressed conditions.

The endocannabinoid system (ECS) has lately been proven to be an important signaling network that modulates a wide range of biological responses in multiple tissues, including the skin, which expresses endocannabinoids such as anandamide, 2-AG, and the endocannabinoid receptor CB1 and/or CB2 [8,20]. Prior studies showed that endocannabinoids could regulate ceramide production via the endocannabinoid receptor CB1-mediated signaling events in certain neuronal cells or cancer cells, e.g., astrocytes, glial cell, glioma cell, as follows [5,8,21]: (i) CB1 activation increases the catalytic action of SMase, which hydrolyze SM to ceramide in the plasma membrane via the factor associated with neutral SMase (FAN)-containing signaling complex-mediated cellular mechanism; (ii) CB1 activation stimulates the activity of de novo synthetic SPT that increases condensation of serine and palmitoyl-CoA to produce ceramide. However, to our knowledge, it has not been identified if the endocannabinoid receptor CB1 is coupled to the generation of ceramide in the skin. We demonstrate here that PS, an analog of the endocannabinoid N-palmitoyl ethanolamine, increases ceramide production by both de novo synthesis and SM hydrolysis pathways to stimulate epidermal barrier integrity under an in vitro AD-like model. Although we first showed the relationship of ECS and ceramide generation in the skin, further studies are still needed to identify the more detailed mechanism(s) of how PS stimulates ceramide production, i.e., the involvement of the FAN-containing signaling complex.

In summary, our studies revealed that the PS-mediated activation of CB1 accounts for the generation of epidermal ceramides through the de novo synthesis and SM hydrolysis pathways. In particular, PS selectively elevates ceramides with 22–26 carbon-containing FAs via activation of CerS2 and CerS3. These findings indicate that PS likely contributes to improving the epidermal barrier function of certain skin diseases, such as atopic dermatitis, in which ceramide levels are dramatically diminished. Hence, pharmacological modulation of CB1 by endocannabinoids, or its analogs, might be considered as therapeutic and/or preventive strategies for such skin conditions.

## 4. Materials and Methods

### 4.1. Cell Culture

Immortalized, non-transformed (HaCaT) human keratinocytes (KC), derived from the human epidermis (a gift from N. Fusenig, Heidelberg, Germany), were grown as described previously [22]. Culture medium was switched to serum-free KC growth medium containing 0.07 mM calcium chloride and growth supplements (Thermo Fisher Scientific, Carlsbad, CA, USA) 1 day prior to the N-palmitoyl serinol (PS, Dr. Raymond Laboratories, Englewood Cliffs, NJ, USA) treatment.

### 4.2. Cell Viability

Cell viability or cytotoxicity was measured by the water-soluble tetrazolium salt (WST) method using the Cell Counting Kit-8 (CCK-8, Dojindo, Japan) in accordance with the manufacturer’s instruction. Briefly, cells (3 × 104 cells/well, 96-well plate) were incubated with PS (10, 25, 50, 100, and 200 μM) for 24 h. 10 μL of the CCK-8 assay kit was then added into each well and the plate was incubated for 1 h prior to reading the absorbance in a microplate reader (Molecular devices M2e, Molecular Devices, Sunnyvale, CA, USA) at 450 nm. The relative cell viability was calculated as the percentage of vehicle-treated cells.

### 4.3. Quantification of Cellular Ceramide Level

To assess the levels of cellular ceramides, human HaCaT KC were pretreated with IL-4 (50 ng/mL) for 20 h, followed by incubation with exogenous PS (25 μM) with or without 10 μM of CB1 inhibitor (AM-251, Tocris Bioscience, Ellisville, MO, USA) for 4 h. Extraction of ceramides was performed as we have reported previously [23]. The extracted lipids were dried using a vacuum system (Vision, Seoul, Korea), re-dissolved in methanol, and analyzed by LC-ESI-MS/MS (API 3200 QTRAP mass, AB/SCIEX, Framingham, MA, USA) in the multiple reaction monitoring (MRM) mode. The ceramide MS/MS transitions (*m/z*) were 510→264 for C14-ceramide, 538→264 for C16-ceramide, 552→264 for C17-ceramide, 566→264 for C18-ceramide, 594→264 for C20-ceramide, 648→264 for C24:1-ceramide, 650→264 for C24-ceramide, 676→264 for C26:1-ceramide, and 678→264 for C26-ceramide, respectively. Data were acquired using the Analyst 1.5.1 software (Applied Biosystems, Foster City, CA, USA). Ceramide levels are expressed in pmol per mg protein.

### 4.4. Enzyme Activity Assay for Sphingomyelinases

Activities of acidic or neutral sphingomyelinases (SMases) were assessed as previously described [23]. Briefly, cells suspended in appropriate SMases assay buffers (acidic SMase buffer: 250 mM sodium acetate, 0.2% Triton X-100, pH 4.5; or neutral SMase buffer: 20 mM HEPES, 0.2% Triton X-100, pH 7.4) were incubated with 5 nmol of C12-sphingomyelin (Avanti Polar Lipids, Alabaster, AL, USA) for 20 min at 37 °C. The reaction was stopped by the addition of CHCl_3_:CH_3_OH (2:1, *v/v*), and the organic phases were dried under N_2_ gas. The residues were then resuspended in MeOH and applied onto the LC-ESI-MS/MS system (API 3200 QTRAP mass, AB/SCIEX, Framingham, MA, USA). The activities of both SMases are expressed as pmol (C12-ceramide) per mg protein per min.

### 4.5. Enzyme Activity Assay for Ceramide Synthases

The ceramide synthases (CerSs) activity assay was performed as described previously [13,24]. Briefly, cell lysates, prepared in an assay buffer (20 mM HEPES, pH 7.4, 25 mM KCl, 2 mM MgCl2, 0.5 mM DTT, 0.1% (*w/v*) fatty acid-free BSA, and 50 μM fatty acid-CoA (C16:0, C18:0, C20:0, C22:0, C24:0, and C26:0)) were incubated with 10 nmol of C17-sphinganine (Avanti Polar Lipids, Alabaster, AL, USA) for 30 min at 37 °C. Total lipids were extracted by the addition of CHCl_3_:MeOH (1:2, *v/v*), and 100 pmol of d17:1/C18:0 ceramide (Avanti Polar Lipids, Alabaster, AL, USA) was used as the internal standard, and applied onto the LC-ESI-MS/MS system (API 3200 QTRAP mass, AB/SCIEX, Framingham, MA, USA), as described previously [24]. The activity of CerS is expressed as pmol (d17:0 dihydroceramide production) per mg protein per min.

### 4.6. Statistical Analyses

All experiments were repeated at least three times. For each experiment, results from triplicate samples were expressed as the mean ± standard deviation (SD). Significance between the groups was determined with an unpaired Student’s *t*-test. The *p* values were set at <0.01.

## Figures and Tables

**Figure 1 ijms-22-08302-f001:**
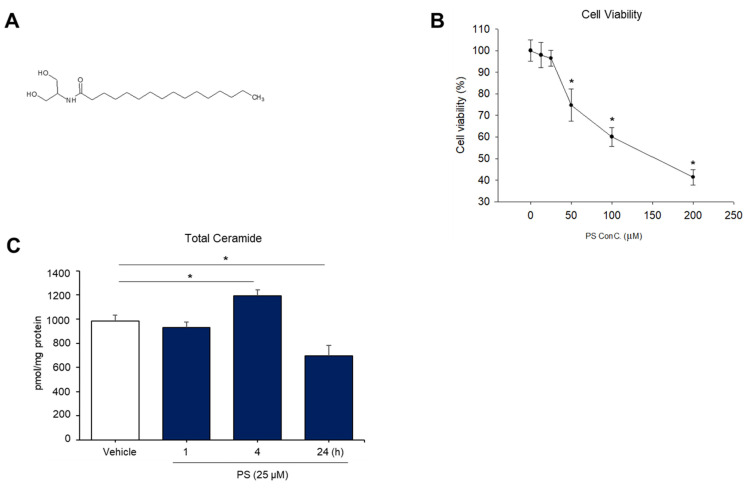
N-palmitoyl serinol (PS)-mediated increase in overall ceramide content in human epidermal keratinocytes (KC). HaCaT KC were incubated with PS (25 μM or as indicated) for up to 24 h. The molecular structure of PS (**A**) and cell viability measured by the WST-1 assay (**B**). Total ceramide content was assessed by the LC-MS/MS system (**C**). Statistical significance was calculated using the unpaired Student’s *t*-test, and significance was defined as * *p* < 0.01 vs. vehicle control (or untreated control).

**Figure 2 ijms-22-08302-f002:**
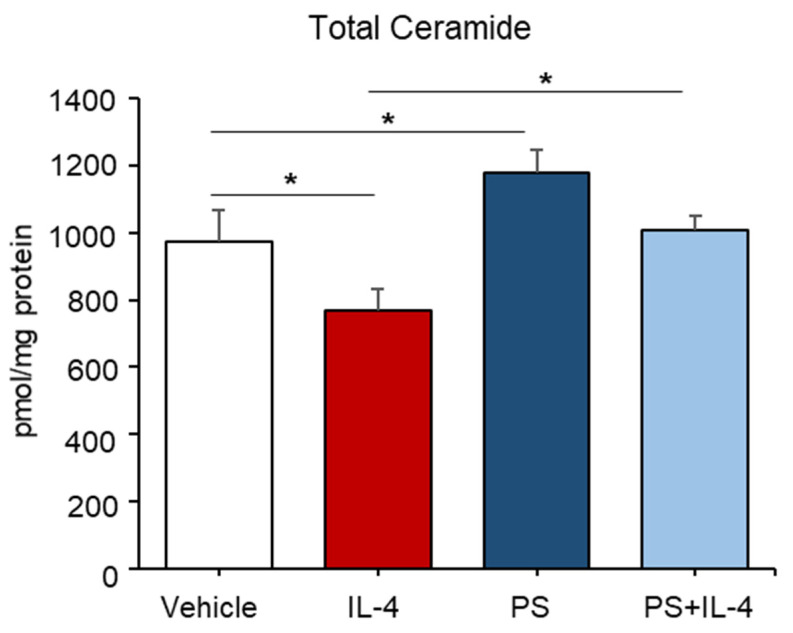
An IL-4-induced decrease in ceramide content was stimulated in cells following N-palmitoyl serinol (PS) treatment. HaCaT KC were pretreated with IL-4 (50 ng/mL) for 20 h, followed by incubation with exogenous PS (25 μM) for 4 h. Total ceramide content was assessed by the LC-MS/MS system. Statistical significance was calculated using the unpaired Student’s *t*-test, and significance was defined as * *p* < 0.01 vs. vehicle control (or untreated control).

**Figure 3 ijms-22-08302-f003:**
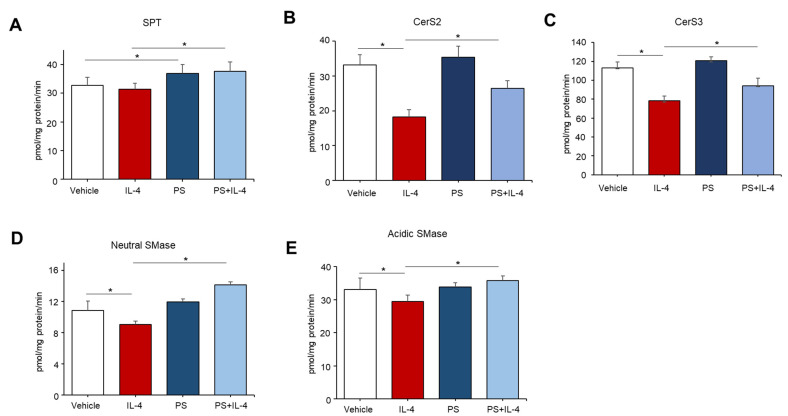
N-palmitoyl serinol (PS)-mediated alteration in activities of key enzymes responsible for ceramide synthesis. HaCaT KC were pretreated with IL-4 (50 ng/mL) for 20 h, followed by incubation with exogenous PS (25 μM) for 4 h. Serine palmitoyltransferase (SPT) (**A**), ceramide synthase (CerS) 2 (**B**), CerS3 (**C**), neutral sphingomyelinase (SMase) (**D**), or acidic sphingomyelinase (SMase) (**E**) were measured by the LC-MS/MS system. Statistical significance was calculated using the unpaired Student’s *t*-test, and significance was defined as * *p* < 0.01 vs. vehicle control (or untreated control).

**Figure 4 ijms-22-08302-f004:**
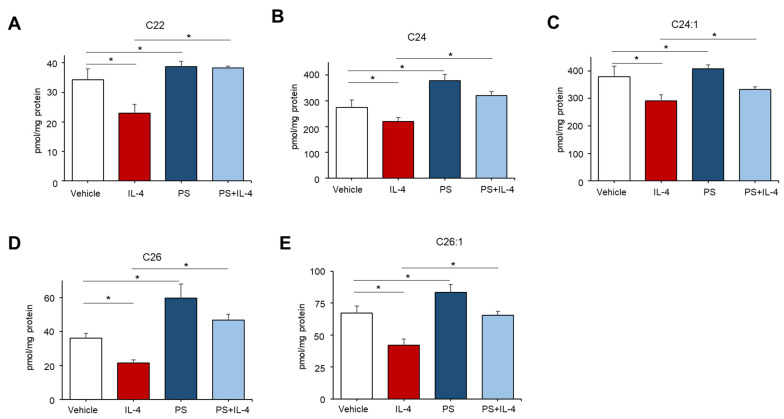
N-palmitoyl serinol (PS) specifically increases ceramides containing long-chain fatty acids (FA). HaCaT KC were pretreated with IL-4 (50 ng/mL) for 20 h, followed by incubation with exogenous PS (25 μM) for 4 h. Ceramides with different carbons in length (C22–C26, (**A**–**E**), respectively) were assessed by the LC-MS/MS system. Statistical significance was calculated using the unpaired Student’s *t*-test, and significance was defined as * *p* < 0.01 vs. vehicle control (or untreated control).

**Figure 5 ijms-22-08302-f005:**
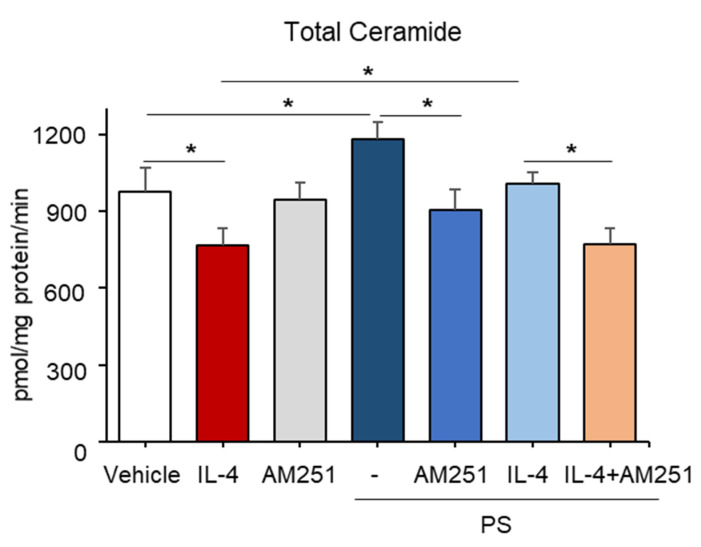
Involvement of the endocannabinoid receptor CB1 in the N-palmitoyl serinol (PS)-mediated stimulation of overall ceramide content. HaCaT KC were pretreated with IL-4 (50 ng/mL) for 20 h, followed by incubation with exogenous PS (25 μM) with or without CB1 inhibitor (AM-251, 10 μM) for 4 h. Total ceramide content was assessed by the LC-MS/MS system. Statistical significance was calculated using the unpaired Student’s *t*-test, and significance was defined as * *p* < 0.01 vs. vehicle control (or untreated control).

**Figure 6 ijms-22-08302-f006:**
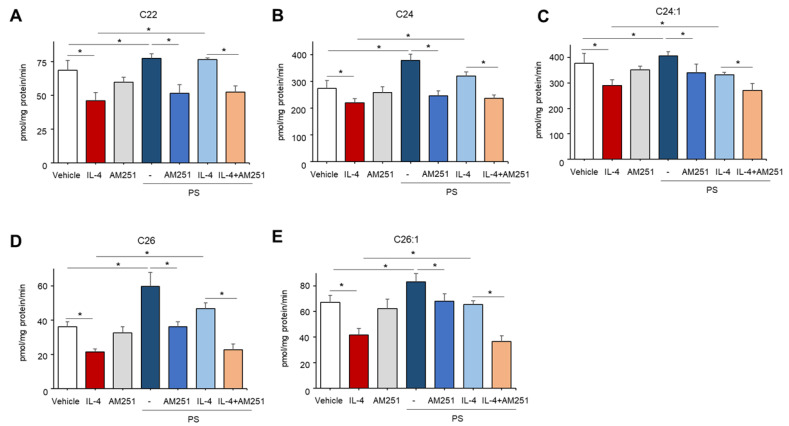
N-palmitoyl serinol (PS) particularly stimulates production of ceramides with long-chain fatty acids (FA) through a CB1-dependent mechanism. HaCaT KC were pretreated with IL-4 (50 ng/mL) for 20 h, followed by incubation with exogenous PS (25 μM) with or without CB1 inhibitor (AM-251, 10 μM) for 4 h. Ceramides with different carbon lengths (C22–C26, (**A**–**E**), respectively) were assessed by the LC-MS/MS system. Statistical significance was calculated using the unpaired Student’s *t*-test, and significance was defined as * *p* < 0.01 vs. vehicle control (or untreated control).

## Data Availability

Not applicable.

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
