# Peer review of "N-Palmitoyl Serinol Stimulates Ceramide Production through a CB1-Dependent Mechanism in In Vitro Model of Skin Inflammation"

_ijms, 2021, doi:10.3390/ijms22158302_

Round 1

Reviewer 1 Report

The study examines the role of production of long chain ceramides in an endocannabinoid receptor, CB1 (an important component of the endocannabinoid system (ECS)) dependent manner to alleviate skin inflammation. Ceramide act as an important lipid component of the skin that acts as a skin barrier to inflammation. The authors successfully here show that palmitoyl serine increases ceramides with long chain fatty acids (C22-C24) through the activation of CerS2 and Cer3 in IL-4 mediated inflamed kerationocytes.  The study was performed very well, all the conclusions are supported by the data, and the article is well written. I only one minor comments.

  1. The authors should also include ceramide data with short chain fatty acids to show that palmitoyl serine selectively increases ceramides with long chain containing fatty acids and not short chain fatty acids.

Author Response

Response to Reviewers

We thank you for your comments and suggestions to improve our study. We respond to each question and comment as follows (Note that changes are highlighted in red).

Reviewer #1: The authors should also include ceramide data with short chain fatty acids to show that palmitoyl serine selectively increases ceramides with long chain containing fatty acids and not short chain fatty acids.

Response: We appreciate your critical comments for improving our study. In response to this reviewer’s comment, we now incorporate additional result (see Suppl Figure 1) showing that overall levels of ceramides including short-chain FAs (≤20) were not altered under these conditions.

Reviewer 2 Report

Well-designed and well-written study which demonstrated that an analog of endocannabinoid, PS stimulates generation of specific ceramide species as well as total amount of ceramides via endocannabinoid receptor CB1-dependent mechanism, resulting
in improvement of epidermal permeability barrier function. The authors highlight that
pharmacological modulation of CB1 by endocannabinoids or its analogues might be considered as therapeutic or preventive strategies for various skin conditions with skin barrier impairement. Congratulations to the authors.

Author Response

We appreciate the reviewer for valuable comments and positive feedback.

Round 2

Reviewer 1 Report

The supplemental data submitted addresses the question asked.